# Comparative Analysis of the Ultrastructure, Bubble Pores, and Composition of Eggshells of Dwarf Layer-White and Guinea Fowl

**DOI:** 10.3390/ani14101496

**Published:** 2024-05-17

**Authors:** Yi-Tong Wang, Yi-Fan Chen, Jun-Jie Zhang, Quan Zhang, Xiao-Yu Zhao, Rong-Yan Zhou, Hui Chen, De-He Wang

**Affiliations:** 1College of Animal Science and Technology, Hebei Agricultural University, Baoding 071001, China; 18340369203@163.com (Y.-T.W.); chenyfchn@163.com (Y.-F.C.); 13514954495@163.com (J.-J.Z.); rongyanzhou@126.com (R.-Y.Z.); 13930259580@163.com (H.C.); 2College of Coastal Agricultural Sciences, Guangdong Ocean University, Zhanjiang 524088, China; zhangg@gdou.edu.cn; 3Baoding Xingrui Agriculture and Animal Husbandry Development Co., Ltd., Baoding 072500, China; 13388088270@163.com

**Keywords:** hen, Guinea fowl, eggshell strength, ultrastructure, bubble pore

## Abstract

**Simple Summary:**

A decrease in eggshell quality in the late laying period leads to eggshell breakage, seriously affecting production efficiency. Guinea fowl with relatively high eggshell strength have the same layers of eggshell structure and various identical proteins as laying hens. However, their eggshell structures have not been systematically compared. Therefore, this study systematically compared the eggshell ultrastructure, bubble pore index, and composition of laying hens and Guinea fowl. The differences in the eggshell ultrastructure, bubble pores, and composition revealed, to some extent, the structure and composition necessary to increase the eggshell quality of laying hens. This study provides a reference for improving eggshell quality.

**Abstract:**

The decrease in eggshell quality seriously affects production efficiency. Guinea fowl (GF) eggs possess strong eggshells because of their unique crystal structure, and few systematic studies have compared laying hen and GF eggs. Sixty eggs were collected from both 40-week-old Dwarf Layer-White (DWL-White) laying hens and GF, and the eggshell quality, ultrastructure, bubble pores, and composition were measured. The results showed that the DWL-White eggs had a higher egg weight and a lower eggshell strength, strength per unit weight, thickness, and ratio than the GF eggs (*p* < 0.01). There were differences in the mammillary layer thickness ratio, the effective layer thickness ratio, the quantity of bubble pores (QBPs), the ratio of the sum of the area of bubble pores to the area of the eggshell in each image (ARBE), and the average area of bubble pores (AABPs) between the DWL-White and GF eggs (*p* < 0.01). The composition analysis demonstrated that there were differences in the organic matter, inorganic matter, calcium, and phosphorus between the DWL-White and GF eggs (*p* < 0.01). There were positive associations between the mammillary knob number in the image and the QBPs and ARBE and a negative correlation with the AABPs in the DWL-White eggs (*p* < 0.01). This study observed distinctions that offer new insights into enhancing eggshell quality.

## 1. Introduction

Eggshell quality is critical in egg production. The probability of egg breakage because of poor eggshell quality is approximately 9–11% [1], which seriously affects the production efficiency of laying hens. The eggshell is formed in the oviduct shell gland, and the formation process is primarily divided into three stages: initial mineralization of the mammillary layer (ML, 70 μm), rapid mineralization of the palisade layer (PL, 200 μm), and terminated calcification of the vertical crystal layer (VCL) and cuticle (10 μm) [2], with durations of approximately 5, 10, and 1.5 h, respectively [3]. The content of eggshells is primarily 94–97% minerals and 3–3.5% organic matter, of which calcium, magnesium, phosphorus, and other mineral elements are primarily in the form of calcium carbonate, magnesium carbonate, calcium phosphate, and other compounds. These compounds account for 94%, 1%, and 1% of the total eggshell mass [4] and play critical roles in the strength, toughness, and elasticity of eggshells [5]. The organic matter of eggshells comprises proteins and proteoglycans, and the proteins account for approximately 2.1% [6]. Guinea fowl (GF) with a relatively high ratio of eggshell strength to egg weight [7] had the same layer of eggshell structure as that of laying hens and had 130 types of identical proteins [8], primarily ovalbumin, lysozyme, ovotransferrin, ocvocleidins, and ocvocalyxins [9]. These proteins can regulate eggshell mineralization by selectively adsorbing minerals and controlling the quantity and morphology of crystals, namely, calcium carbonate [9]. For example, Ovocleidin-116 can inhibit calcium carbonate precipitation [10] and transform the crystal spiral growth mechanism into an aggregation growth mechanism [9]. Additionally, a study found no difference in the total protein content between high- and low-strength eggshells and that the contents of ovalbumin and ovotransferrin decreased [11].

Guo et al. [12] showed that eggshell strength (ESS) is a medium heritability trait, with a size of approximately 0.26–0.43. Other studies have shown that ESS primarily depends on the thickness and ultrastructure of the eggshell [13,14], in which the correlation coefficient between the ESS and eggshell thickness (EST) was 0.45–0.47 [15,16] and the correlation coefficient with the PL thickness was 0.51–0.61 [14,17]. The body weight and egg weight (EW) of GF are similar to those of laying hens; however, the ESS of GF can reach 2.47 times that of laying hens [18]. Additionally, the ratio of the ESS to the EW of GF was the highest among 13 species of birds, including chickens, ducks, geese, eagles, and ostriches [7,19], probably because the ultrastructure of GF and the other species differed. Under polarized light microscopy, most eggshell crystals grow vertically along the c-axis into adjacent columnar structures from the inside to the outside [13]. The GF eggshell also showed a structure similar to that of the ML. From the PL, a complex interlaced structure was formed [20] by many small and disoriented crystals [18]. Rodriguez-Navarro et al. [13] showed that the correlation coefficient between the crystal orientation randomness of eggshells and the ESS was 0.63, and the crystal selection of the randomness of GF eggshells was higher than that of laying hens [21]. Additionally, in GF eggshells, the genetic correlation coefficients of the crystal size and orientation with the mammillary layer thickness were 0.65 and 0.66, respectively [22]. This crystal orientation may improve the ESS by improving the ultrastructure of the ML or PL. In addition, studies have shown that eggshell color also affects eggshell strength to a certain extent. Both Drabik et al. [23] and Yang et al. [24] showed that the darker eggshell color had higher strength. At the same time, Drabik et al. [23] also observed the eggshell structure of different eggshell colors. They found that the light eggshell would have some microcracks, while the dark eggshell was relatively uniform.

Avian eggshell porosity plays a crucial role in gas exchange during egg incubation [25], but its effect on the eggshell microstructure and quality remains unclear. Studies of avian eggshell porosity have primarily focused on gas pores in eggshells. For example, Tyler et al. [26] explored the distribution law of gas pores on the surface of eggshells, obtained a non-uniform distribution similar to a negative binomial distribution, and preliminarily discussed the relationship between pores and ESS. Other researchers have divided or classified pores into five types [27] and seven types [28] according to pore location or morphology. Using new technology, Zhou et al. [29] systematically studied eggshell porosity by using scanning electron microscopy and demonstrated three levels: nanoscale pores with a diameter of <10 nm between the VCL and cuticle, bubble pores with a diameter of approximately 250 nm dispersed in the mineralized eggshell layer, and pores with a diameter of several microns throughout the mineralized eggshell layer. Gas penetration experiments have shown that the eggshell primarily regulates gas conduction by adjusting the number and size of bubble pores rather than gas pores. Therefore, bubble pores may play an important role in eggshell porosity. The formation of gas pores is primarily attributed to the incomplete fusion of the early mammillary cores and columnar extension in the PL [30]. Notably, according to a review of the literature, the formation mechanism and formation rules of bubble pores have not been explored, but they may be critical in understanding the microstructures of eggshells.

Currently, the research on eggshell structure focuses on the comparison of Galliformes and Anseriformes [31], GF and Graylag goose [21], and a different breed [32], but the eggshell structure of laying hens and GF of Galliformes has not been systematically compared in the literature. In addition, laying hens and GF have hybridized to produce offspring [33]. Thus, to provide novel insights into improving eggshell quality, this study aimed to explore the relationships among eggshell quality, composition, and bubble pores by comparing the differences in the ultrastructure, composition, bubble pore structure, and distribution between laying hens and GF.

## 2. Materials and Methods

### 2.1. Experimental Hens and GF

The Dwarf Layer-White (DWL-White) laying hens and GF used in this experiment were 40 weeks old. The 1200 DWL-White hens were raised at the College of Animal Science and Technology of the Hebei Agricultural University (Baoding, China). Their line was selected from Dwarf Brown-shell layers and bred from the progressive hybridization of Nongda Brown and French Star broilers developed in 1995 [34]. Their average mature weight was approximately 1500 g, their production started at 22 weeks old, and the egg production rate at 40 weeks old was 82%. The 2000 GF were raised at the College of Coastal Agricultural Sciences, Guangdong Ocean University (Zhanjiang, China). The average mature weight of the GF used in this study was approximately 1900 g, production started at 26 weeks old, and the egg production rate was 80%. The two strains were reared in a fully enclosed house in individual cages with free access to food and water, a 16:8 h light–dark cycle, and automatic control of the mechanical ventilation. The basal diet during the experiment was based on the Feeding Standard for Chickens (NY/T33-2004) [35].

### 2.2. Egg Quality Measurement

We collected 60 eggs produced on the same day from 40-week-old DWL-White in the same feeding environment. The GF eggs were collected in the same way. The initial inclusion criteria for the eggs were that each egg had a smooth surface; no feces, blood, or breakage; and completed the measurement of eggshell quality. The egg quality indicators were the egg long length (ELL), egg short length (ESL), eggshell index (ESI), eggshell surface area (ESA), EW, ESS, eggshell weight, strength per unit weigh (SUW), EST, eggshell membrane thickness (ESMT), and eggshell ratio (ESR). The GF ESS was determined using a Texture analyzer (TA-XT2i, Stable Micro Systerms, Godalming, UK). The DWL-White ESS was determined using an ESS tester (ESTG-01; Israel Aoke Co., Ltd., Tel Aviv, Israel). The ESA is based on the egg weight formula K × EW^2/3^, where K has a value of 4.67 for an EW < 60 g, 4.68 for an EW between 60 and 70 g, and 4.69 for an EW > 70 g [36]; the SUW is the ratio of the ESS to EW; and the ESMT is the value of the total EST minus the thickness of the mineralized eggshell layer. For other measurement methods of eggshell quality indicators, please refer to our previous study [37].

### 2.3. Measurement of Ultrastructure and Bubble Pores of Eggshell

After measuring the egg quality indices, 30 eggs each from the DWL-White hens and GF with a relatively complete eggshell structure were randomly selected. Two samples with areas of approximately 1 × 1 cm^2^ were collected from the blunt, middle, and sharp ends of the eggshell. One sample was used to measure the ultrastructure and bubble pore indexes of different mineralized layers in the cross-section, and the other sample was used to observe the related mammillary knobs on the inner surface of the eggshell. For the collected eggshell samples, the surface stains of the eggshells were first washed with distilled water; next, the eggshells were boiled with 1%NaOH solution (Tianjin Damao Chemical Reagent Factory, Tianjin, China) for 15 min to remove the eggshell membrane. A light microscope (13395H2X; Leica Microsystems Inc., Buffalo, Wetzlar, Germany) was used to ensure that the eggshell membrane fibers on the mineralized eggshell papilla were completely removed. The NaOH on the eggshell was washed with distilled water, and the eggshell sample was left at room temperature for 2 h to dry naturally.

Before measuring the ultrastructure and bubble pores of the eggshell, the dried eggshell sample was coated with gold for 45 s by using an ion sputter (Sputter Coater108, Cressington Scientific Instruments Ltd., Watford, UK); the aim was to increase the electrical conductivity. Subsequently, scanning electron microscopy (Prisma E, EI Segundo, CA, USA, Thermo Fisher Scientific, Waltham, MA, USA; vacuum condition is 10^−3^ pa) was used to capture images of the mammillary knobs inside the eggshell at 200× magnification and the ultrastructure of the mineralized layer across the eggshell at 400× magnification. Bubble pores in the ML, PL, and VCL were imaged at 10,000× magnification. For images captured at different magnifications, the number of mineralized layer mammillary knobs was manually labeled using Photoshop software (Photoshop CC 2018, Adobe Systems Corp., San Jose, CA, USA). The mammillary knob number in the image (MAN) was counted using the taskbar count tool, the line tool was used to manually divide the boundary between the eggshell ML and PL, and the bubble pores were “painted” black to increase their color contrast with the remaining eggshell. Finally, the contours of each bubble pore were automatically identified and extracted using ImageJ Pro Plus software (version 6.0, Media Cybernetics Corp., Silver Spring, MD, USA).

The thickness of the different mineralized layers across the eggshell was measured using Image J software (version 1.41, National Institutes of Health, Bethesda, MD, USA). The quantity of the bubble pores (QBPs); the average area of the bubble pores (AABPs); the sum of the area of the bubble pores (ASBPs); the ratio of the sum of the area of the bubble pores to the area of the eggshell in each image (ARBE); the average perimeter of the bubble pores (APBPs); and the area of the eggshell in each image (AEEI) on the ML, PL, and VCL were also measured using ImageJ Pro Plus software. Because the boundaries of the PL, VCL, and cuticle are difficult to distinguish under a scanning electron microscope, their thicknesses are called the effective layer. The mammary layer thickness ratio (MTR) and effective layer thickness ratio (ETR) were calculated using the mammillary layer thickness (MT) and the effective layer thickness (ET). The specific measurement methods for each index of the eggshell bubble pores were based on those of our prior research [38].

### 2.4. Eggshell Chemical Components Measurement

After measuring the eggshell quality, 10 eggshells were randomly selected from the two poultry strains, boiled in 1% NaOH solution for 15 min to remove the eggshell membrane, and rinsed with distilled water and dried naturally for 2 h. Next, each eggshell was ground in a mortar for 20 min to powder, and 1.0 g eggshell powder was measured using a balance with an accuracy of 0.001 g (PWN125DZH, OHAUS, Parsippany, NJ, USA) and placed in a crucible (crucible constant weight M0). The 1.0 g sample was baked in an electric blast drying oven (GZX-9240MBE, Shanghai Boxun Industrial Co., Ltd., Shanghai, China) at 110 °C for 5 h; removed; cooled in a desiccator for 30 min; weighed; baked for 1 h; cooled; and repeatedly weighed until the difference between the two weights was <0.002 g, obtaining weight M1. Subsequently, using an electric furnace (DL-1, Beijing ever briGht Medical Treatment Instrument Co., Ltd., Beijing, China) to char the 1 g sample until a smokeless condition was observed, it was transferred to a high-temperature furnace (JC-MF-7A, Qingdao Juchuang Environmental Protection Group Co., Ltd., Qingdao, China) and burned at 550 °C for 12 h to obtain crude ash. Next, the 1 g sample was cooled in the dryer for 30 min, and M2 was obtained. The percentage of organic matter in each eggshell sample was calculated using the formula:organic matter (%) = (M2 − M0)/(M1 − M0) × 100%

The potassium permanganate titration method was used to measure the calcium content in the sample [39], using the formula:Ca (%) = [(V − V0) × c (1/5 KMnO_4_) × M(1/2Ca)]/10 × m × V1

The spectrophotometry method was used to measure the phosphorus content in the sample [39], using the formula:P (%) = X/m × 10^5^ × V

Atomic Absorption Spectroscopy was used to determine the content of magnesium in the samples [39]. Magnesium standard liquid (10 μg/mL) of 0 mL, 2.0 mL, 4.0 mL, 6.0 mL, 8.0 mL, and 10.0 mL was placed in a 100 mL volumetric bottle, 5 mL lanthanum chloride solution was added, and the volume was diluted to the scale with deionized water to prepare a standard series concentration of 0 μg/mL, 0.1 μg/mL, 0.2 μg/mL, 0.4 μg/mL, 0.6 μg/mL, 0.8 μg/mL, and 1.0 μg/mL. The concentration was then determined using an atomic absorption spectrometer (Xplor AA, GBC Scientific Equipment Ltd., Braeside, Victoria, Australia).

### 2.5. Statistical Analysis

The data of the egg quality, eggshell ultrastructure, and bubble pores were removed by using mean ± 3 SD, and the measurement values of the organic matter, inorganic matter, calcium, phosphorus, and other elements were removed using the quartile method. The differences in each index between the DWL-White hens and GF were then analyzed by conducting an independent sample *t* test in SPSS (IBM SPSS Statistics version 25.0, Armonk, NY, USA); the differences among the eggshell ML, PL, and VCL within the group and between the blunt, middle, and sharp ends of the eggshell were tested using one-way analysis of variance; and the differences among the groups were tested using Duncan’s multiple comparison. The relationship between the eggshell quality and ultrastructure was analyzed using Pearson’s correlation coefficient.

## 3. Results and Discussion

### 3.1. Eggshell Quality

Table 1 shows the differences in the eggshell quality between the DWL-White hens and GF. Regarding eggshell quality, the ELL, ESL, ESA, and EW of the DWL-White hens were significantly higher than those of the GF (*p* < 0.01). The ELL, ESL, and ESA reflect the egg size, determined by the EW as a whole, and the egg weight of laying hens of the same breed is positively correlated with weight [40]. In this study, the EW of the DWL-White hens was higher than that of the GF, which may be attributed to the increase in the EW of the commercial laying hens represented by the DWL-White hens via systematic artificial selection [34,41]. Additionally, the ESS, EST, ESR, SUW, and ESMT of the DWL-White hens were significantly lower than those of the GF (*p* < 0.01), indicating that the mechanical properties of the GF eggshells were better than those of the DWL-White hens, consistent with Petersen and Tyler’s [18] description that the ESS of GF was higher than that of laying hens. The correlation coefficient between the EST and ESS was 0.33–0.47 [15,16,42], and the correlation coefficient with the ESR was 051–0.74 [43]. Therefore, EST is an important factor that affects the ESS and ESR.

SUW refers to the ESS per unit weight, which excludes the influence of the EST to a certain extent; thus, the understanding of the influence of the eggshell ultrastructure or internal crystal characteristics on ESS could be improved. Studies on the microstructure of eggshells have shown that the formation of calcium carbonate crystals in eggshells plays an important role in ESS [13,44,45,46]. Dunn et al. [22] showed that the size, shape, and orientation of calcium carbonate crystals in eggshells are particularly important for their mechanical properties. Therefore, the higher SUW value in the GF than that in the DWL-White hens may have resulted from the crystal structure inside the eggshell of the former being superior to that of the latter. The initial mineralization phenotype of eggshells may play an important role in the structure and strength of eggshells [47], and the eggshell membrane, as a platform for the initial mineralization of eggshells, plays an important role in the calcification and nucleation of eggshell mammillaries [3]. However, the underlying mechanisms remain unclear. In this study, the ESMT of the GF was significantly higher than that of the DWL-White hens, suggesting that regulating the initial mineralization of the eggshell would affect the eggshell ultrastructure or ESS.

### 3.2. Eggshell Ultrastructure and Bubble Pores

The eggshells of the DWL-White hens and GF had similar ultrastructures (Figure 1 and Figure 2), both of which contained the ML and an effective layer with obvious phenotypic characteristics. Table 2 shows the quantitative differences in the eggshell ultrastructure and bubble pore-related indexes between the DWL-White hens and GF. As shown in Table 2, the MTR of the DWL-White was 41.99%, significantly higher than that of the GF (*p* < 0.01), and the ETR was 58.01%, significantly lower than that of the GF (67.06%) (*p* < 0.01). The difference in the MTR between the DWL-White and GF may be caused by structural differences in the ML. Although the MLs in both strains were composed of round particles aggregated along the c-axis to form adjacent columnar structures [21,48], the columnar crystals of the GF were more compact and the width of the columnar structure was narrower than those of the laying hens [20,47].

In another study related to poultry, Tang [49] reported that the ETR of chicken, duck, and goose eggshells was approximately 53.41%, 59.39%, and 60.18%, respectively, all lower than the ETR of the GF in this study and similar to the measured value of eggshells. These results suggest that the ETR and ET play important roles in ESS to some extent. Additionally, the crystal structure of the PL differed between the laying hens and GF. The literature has shown that the crystal characteristics of the PL of GF formed are not obvious, and the adjacent crystals are interleaving [20]; however, the crystal orientation of the eggshell PL of laying hens is relatively single [13], forming a diameter (width) of 70–100 μm [13,22,48] and a relatively independent columnar structure. Rodríguez-Navarro et al. [50] observed the eggshells of GF at different mineralization stages, and they found organic-rich structures (mammalian cores) protruding from the outer surface of the eggshell membrane that were similar to those found [51] on laying hens. Although the initial mineralized structures of the two were similar, the eggshell strength showed significant differences. This shows that the orientation of the crystal inside the eggshell plays a vital role in the strength of the eggshell. Combined with the higher SUW measurements of the GF in this study, the higher ESS is likely owing to its good crystal structure. As shown in Table 2, the QBPs, ASBPs, and ARBE of the DWL-White were significantly higher than those of the GF (*p* < 0.01), and the AABPs and APBPs were significantly lower than those of the GF (*p* < 0.01), indicating to the characteristics of many small bubble pores in the DWL-White and a few large bubble pores in the GF (Figure 3). Bubble pores are important pores widely distributed throughout the eggshell. In the literature, scanning electron microscopy showed that such pores accounted for 3.8–4.4% of the eggshell volume of laying hens [38], consistent with the results of this study. Additionally, bubble pores may play a more important role than the gas pores that permeate the eggshell in regulating the internal and external gas exchange of the egg [25]. Damaziak et al. [52] showed that the hatching rate of GF reached 89.47%, indicating that few but large pores may facilitate the gas exchange inside and outside the eggshell during embryo hatching. Fathi et al. [32] also observed the bubble pore structure in the PL. Most of them are arranged in straight lines forming tiny voids, like a spongy appearance. These bubble pores may help reduce the eggshell weight.

Few studies have investigated the formation of bubble pores in eggshells, and the formation mechanism remains unclear. According to Jonchère [53], the Ca^2+^, HCO_3_^−^, and CO_3_^2−^ plasma in the uterine fluid, gas, and water produced during the formation of eggshells may lead to the formation of bubble pores in the crystal, or H^+^ may cause the dissolution and re-precipitation crystallization of calcium carbonate crystals [54], leading to the formation of bubble pores. Additionally, bubble pores may also be the result of the normal growth and deposition of calcium carbonate crystals in eggshells, regulated by the matrix proteins of uterine fluid [48], and calcium carbonate in eggshells exists in the form of irregular hexahedra, such as 104, 108, 110, 113, and 202 [22]. The significant difference in the crystal structure of the ML and PL between the DWL-White and GF [20] may have led to differences in the number and size of the bubble pores between the two strains. Additionally, the appearance of bubble pores may affect the ESS to a certain extent. For example, in metal matrix composites, the tensile strength decreases with increasing porosity content within 1–7% of the porosity content [55], which seriously affects the mechanical properties of materials.

### 3.3. Eggshell Bubble Pore

Table 3 shows the differences in the indicators of bubble pores among the different eggshell layers and ends in the DWL-White and GF eggshells. In the DWL-White, there were significant differences in the QBPs, ASBPs, and ARBE among the different mineralized layers (*p* < 0.05), in which the average values of each index in the PL were the highest, followed by the VCL and ML; additionally, the AABPs of the ML was significantly lower than that of the PL (*p* < 0.05). In the GF, the QBPs, ASBPs, and ARBE in the ML were significantly lower than those in the PL and VCL (*p* < 0.05), the ASBPs and ARBE in the PL and VCL had no significant difference, and the QBPs in the VCL tended to be lower than that in the PL (*p* > 0.05). Additionally, the AABPs of the VCL was significantly higher than that of the ML and PL (*p* < 0.05). The results of this experiment support those of Zhou et al. [29] and Arzate-Vazquez et al. [25]. In conclusion, the quantity, size, and sum of the area of bubble pores increased and then decreased with the mineralization and deposition of calcium carbonate during the formation of the DWL-White medium eggshell. The variation law of the GF differed from that of the DWL-White, primarily reflected in the gradual increase in the bubble pore size during the mineralization process from the PL to the VCL, resulting in no obvious decrease in the ASBPs. The difference in the bubble pores between the two strains may be because of the higher overall bubble pore ratio of the DWL-White eggs and the lower bubble pore ratio of the GF eggs.

There were no significant differences between the QBPs, AABPs, and ASBPs at the blunt, medium, and sharp ends of the DWL-White and GF (*p* > 0.05); however, the ASBPs and ARBE at the blunt end were higher than those at the middle and sharp ends. During eggshell formation, the type and amount of matrix proteins in the uterine fluid change continuously with the deposition of mineralized materials [56,57], which may lead to changes in the type and size of calcium carbonate crystals [22], causing changes in a series of the indicators of bubble pores. The variation in each index of the blunt, medium, and sharp bubble pores in eggshells can be affected primarily by the geometric shape and radius of the different ends.

### 3.4. Eggshell Chemical Components

Table 4 shows the differences in the eggshell chemical components between the DWL-White and GF. The organic matter content of the DWL-White was significantly lower than that of the GF, and the inorganic matter content was significantly higher than that of the GF (*p* < 0.01). In the inorganic components, the content of calcium was significantly lower than that of the GF (*p* < 0.05) and the content of phosphorus was significantly higher than that of the GF (*p* < 0.01). The literature has reported that there are 149 proteins in the eggshell matrix of GF [8], among which Dromaiocalcin-1-like, Ovocleidin-17-like, and Extracellular Fatty Acid-Binding protein X2 are the main proteins, with relative contents of 4231-80911, 2771-14838, and 2006-14499 emPAI, respectively. However, in the literature, there were 466–675 matrix proteins in laying hens [56,58,59], among which lysozyme and ovotransferrin were relatively abundant, with relative contents of 9999.9–12894 and 928.57–1482.5 emPAI, respectively. There were fewer protein types in the eggshell matrix of the GF than in the DWL-White, but the average protein content was higher than that of the DWL-White laying hens, which reflected the germplasm specificity between the two strains and explained the difference in organic matter to a certain extent.

Calcium is the most abundant mineral element in eggshells; it combines with CO_3_^2−^ to form calcium carbonate crystals and forms the structural basis of eggshells. The absolute and relative contents of calcium ions in the eggshell composition of the GF were higher than those in the DWL-White laying hens, resulting in the relative contents of phosphorus, magnesium, and other elements to a certain extent. Magnesium in eggshells usually exists as carbonates, which can catalyze and regulate the crystal formation and contribute to the formation of solid and complete crystal structures. Matrix proteins such as ovotransferrin, ovocleidins, and ovocalyxins play a more important role in regulating the morphology of calcium carbonate crystals than inorganic substances [47,57,59], suggesting that the special crystal structure and higher ESS in GF than in laying hens is regulated primarily by matrix proteins, such as ovotransferrin. Phosphorus is primarily present in the VCL of eggshells in the form of hydroxyapatite crystals [3,34,60], and an increase in the phosphorus concentration in eggshells marks the beginning of eggshell termination calcification [61].

### 3.5. Correlations

Table 5 describes the phenotypic correlations between the eggshell quality and ultrastructural indicators in the two strains. In both groups, the ESS positively correlated with the EST and ET, the EST positively correlated with the MT and ET (*p* < 0.05), and the MT negatively correlated with the MAN (*p* < 0.05). ESS is primarily affected by the EST, in addition to the morphology of eggshell calcium carbonate crystals [14], and the effective layer, as the main structure of mineralized eggshell layers, plays an important role in improving the ESS; therefore, the ESS, EST, and ET are strongly positively correlated. There was a significant negative correlation between the MT and MAN, consistent with the findings in Duan [62]. This result may be because the dense mammillary knob number leads to the early fusion of the ML, affecting MT growth. In the DWL-White, there was a significant negative correlation between the MAN and AABPs (*p* < 0.05) and a significant positive correlation between the MAN and QBPs, ASBPs, and ARBE *(p* < 0.05). However, there was no correlation between the MAN and the various indices of the bubble pores in the GF (*p* < 0.05). Additionally, the QBPs negatively correlated with the AABPs (*p* < 0.05) and positively correlated with the ASBPs and ARBE (*p* < 0.05) in both strains.

In DWL-White, the denser mammillary knobs produce smaller calcium carbonate crystals during the growth and fusion process [16], which may increase the production of CO_2_ and other gases or the dissolution of H^+^ [53,54], forming a lot of QBPs. In GF, the formation pattern differs from that of laying hens because of the few overall QBPs. The negative correlation between QBPs and AABPs may play a certain role in maintaining the balance of eggshell porosity, and the positive correlation between QBPs and ASBPs and ARBE indicates that avian may be more inclined to regulate porosity by regulating the QBPs than by regulating the diameter of bubble pores. The correlation between the ESS and EST in the mineralized layers of the DWL-White and GF was similar, but there were differences in the relationship between the bubble pore indicators and the ESS and EST in the eggshells.

## 4. Conclusions

In this study, the ultrastructure, bubble pore indicators, and composition of eggshells of laying hens and GF were systematically compared for the first time. The results showed that the two species had certain similarities in eggshell structure and composition. Specifically, in the ultrastructure, the relative thickness of the effective layer of the laying hens was significantly lower than that of the GF; the QBPs in the laying hens was significantly higher than that of the GF; and the AABPs was significantly lower than that of the GF. In the composition, the calcium content of the laying hens was significantly lower than that of the GF, and the phosphorus content was significantly higher than that of the GF. The difference in each indicator reveals the structure and composition basis of the high eggshell quality of GF to a certain extent, and this study provides a reference for improving the eggshell quality.

## Figures and Tables

**Figure 1 animals-14-01496-f001:**
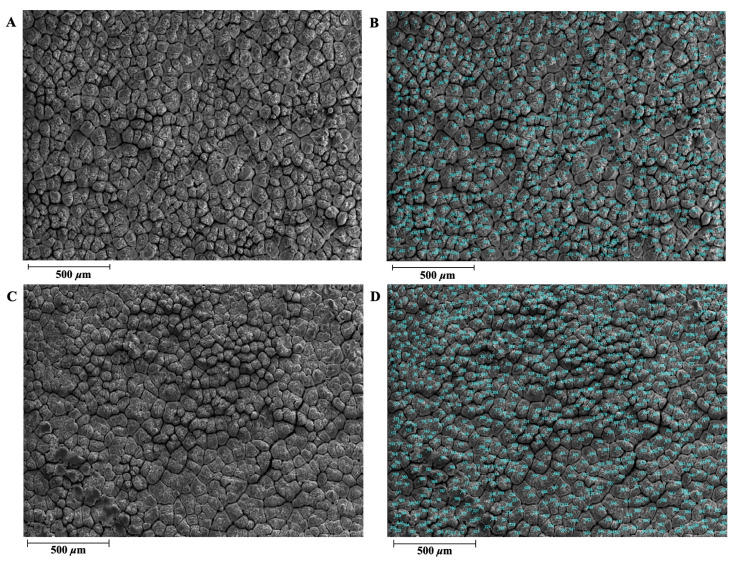
Illustration of Dwarf Layer-White and Guinea fowl eggshell mammillary ultrastructure traits. (**A**) Dwarf Layer-White eggshell mammillary ultrastructure. (**B**) Mammillary knobs of Dwarf Layer-White were manually counted using Photoshop software (Photoshop CC 2018, Adobe Systems Corp., San Jose, CA, USA). (**C**) Guinea fowl eggshell mammillary ultrastructure. (**D**) Mammillary knobs of Guinea fowl were manually counted using Photoshop software (Photoshop CC 2018, Adobe Systems Corp., San Jose, CA, USA).

**Figure 2 animals-14-01496-f002:**
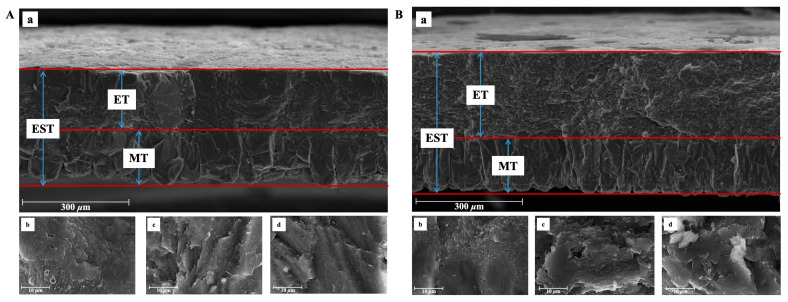
Illustration of Dwarf Layer-White (**A**) and Guinea fowl (**B**) eggshell ultrastructure traits. (**a**) Cross-section structure of eggshell. EST, ET, and MT represent eggshell thickness, effective layer thickness, and mammillary layer thickness, respectively. And ET is the combined thickness of the palisade, vertical crystal, and cuticle layer; (**b**) structure of eggshell mammillary layer; (**c**): structure of eggshell palisade layer; and (**d**): structure of eggshell vertical crystal layer.

**Figure 3 animals-14-01496-f003:**
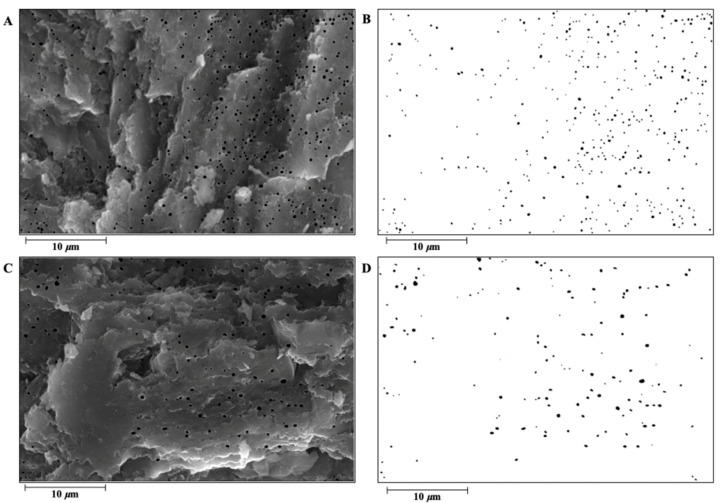
An illustration of the bubble pore structure in the Dwarf Layer-White and Guinea fowl eggshell palisade layer. (**A**) The bubble pores of the Dwarf Layer-White were manually “painted” black using Photoshop software (Photoshop CC 2018, Adobe Systems Corp., San Jose, CA, USA) to increase the color contrast with other areas of the eggshell; (**B**) the bubble pore traits of the Dwarf Layer-White were measured by using Image J software (version 1.41, National Institutes of Health, Bethesda, MD, USA). (**C**) The bubble pores of the Guinea fowl were manually “painted” black using Photoshop software (Photoshop CC 2018, Adobe Systems Corp., San Jose, CA, USA) to increase the color contrast with other areas of the eggshell; (**D**) the bubble pore traits of the Guinea fowl were measured by using Image J software (version 1.41, National Institutes of Health, Bethesda, MD, USA).

**Table 1 animals-14-01496-t001:** Differences in eggshell quality between Dwarf Layer-White and Guinea fowl.

Traits	Dwarf Layer-White	Guinea Fowl
ELL (cm)	5.48 ± 0.19 ^a^	5.30 ± 0.21 ^b^
ESL (cm)	4.19 ± 0.12 ^a^	4.05 ± 0.13 ^b^
ESI	1.31 ± 0.47	1.31 ± 0.40
ESA (cm^2^)	65.80 ± 3.08 ^a^	62.82 ± 3.02 ^b^
EW (g)	52.76 ± 3.71 ^a^	49.22 ± 3.57 ^b^
ESS (kg/cm^2^)	3.38 ± 0.78 ^b^	9.70 ± 3.28 ^a^
SUW	0.06 ± 0.01 ^b^	0.20 ± 0.07 ^a^
ESMT (μm)	11.66 ± 7.44 ^b^	16.94 ± 10.66 ^a^
EST (μm)	302.09 ± 54.92 ^b^	458.78 ± 55.97 ^a^
ESR (%)	12.73 ± 1.66 ^b^	17.93 ± 1.63 ^a^

In the same row, values with no letter mean no significant difference (*p* > 0.05), while values with different lowercase letter superscripts indicate significant differences (*p* < 0.01). ELL = egg long length; ESL = egg short length; ESI = eggshell index; ESA = eggshell surface area; EW = egg weight; ESS = eggshell strength; SUW = eggshell strength per unit weigh; ESMT = eggshell membrane thickness; EST = eggshell thickness; ESR = eggshell ratio; SUW = ESS/EW; ESI = ELL/ESL; ESR (%) = ESW/EW × 100; and ESA = 4.67 × EW^2/3^.

**Table 2 animals-14-01496-t002:** Differences in eggshell ultrastructure of Dwarf Layer-White and Guinea fowl.

Traits	Dwarf Layer-White	Guinea Fowl
MAN	787.03 ± 93.32	801.83 ± 115.54
MTR (%)	41.99 ± 3.81 ^a^	32.72 ± 3.36 ^b^
ETR (%)	58.01 ± 3.81 ^b^	67.28 ± 3.36 ^a^
QBPs	687.51 ± 79.42 ^a^	20.79 ± 11.51 ^b^
AABPs (μm^2^)	0.05 ± 0.01 ^b^	0.15 ± 0.09 ^a^
ASBPs (μm^2^)	37.15 ± 5.92 ^a^	1.54 ± 0.42 ^b^
ARBE (%)	3.23 ± 0.47 ^a^	0.14 ± 0.04 ^b^
APBPs (μm)	0.84 ± 0.04 ^b^	1.09 ± 0.51 ^a^

In the same row, values with no letter mean no significant difference (*p* > 0.05), while values with different lowercase letter superscripts indicate significant differences (*p* < 0.01). MAN = mammillary knob number in image; MTR = mammillary layer thickness ratio; ETR = effective layer thickness ratio; QBPs = quantity of bubble pores; AABPs = average areas of bubble pores; ASBPs = sum of the area of bubble pores; ARBE = ASBPs/the area of the eggshell in each image; and APBPs = average perimeter of bubble pores.

**Table 3 animals-14-01496-t003:** Differences in indicators of bubble pore among different eggshell layers and ends in Dwarf Layer-White and Guinea fowl eggshell.

	Traits	Mean Value of 3 Eggshell Layers			Mean Value of 3 Eggshell Ends		
		ML	PL	VCL	Blunt End	Middle End	Sharp End
Dwarf Layer-White	QBPs	532.06 ± 151.20 ^c^	1236.55 ± 233.77 ^a^	819.35 ± 122.37 ^b^	840.29 ± 311.40	905.08 ± 410.53	798.03 ± 320.55
	AABPs (μm^2^)	0.058 ± 0.011 ^b^	0.067 ± 0.011 ^a^	0.064 ± 0.104 ^ab^	0.068 ± 0.019	0.060 ± 0.017	0.062 ± 0.021
	ASBPs (μm^2^)	31.93 ± 14.24 ^c^	82.31 ± 14.00 ^a^	53.00 ± 18.73 ^b^	61.51 ± 27.54	51.36 ± 24.53	50.53 ± 25.33
	ARBE (%)	2.89 ± 1.39 ^c^	7.20 ± 1.23 ^a^	4.64 ± 1.64 ^b^	5.38 ± 2.41	4.50 ± 2.15	4.42 ± 2.22
	APBPs (μm)	0.90 ± 0.09	0.96 ± 0.13	0.95 ± 0.12	0.99 ± 0.18	0.91 ± 0.17	0.90 ± 0.28
Guinea fowl	QBPs	5.96 ± 3.05 ^b^	19.59 ± 10.48 ^a^	14.73 ± 18.93 ^a^	15.32 ± 9.49	12.64 ± 6.38	11.1 ± 6.87
	AABPs (μm^2^)	0.086 ± 0.052 ^b^	0.058 ± 0.046 ^b^	0.122 ± 0.064 ^a^	0.096 ± 0.076	0.079 ± 0.055	0.106 ± 0.070
	ASBPs (μm^2^)	0.49 ± 0.30 ^b^	1.25 ± 0.69 ^a^	1.24 ± 0.55 ^a^	1.13 ± 0.64	0.92 ± 0.75	0.94 ± 0.65
	ARBE (%)	0.06 ± 0.04 ^b^	0.11 ± 0.06 ^a^	0.12 ± 0.06 ^a^	0.12 ± 0.09	0.08 ± 0.64	0.09 ± 0.08
	APBPs (μm)	0.98 ± 0.31	0.83 ± 0.40	0.98 ± 0.33	0.91 ± 0.53	0.84 ± 0.39	1.09 ± 0.48

In the same row, values with no letter or the same letter superscripts mean no significant difference (*p* > 0.05), while values with different lowercase letter superscripts indicate significant differences (*p* < 0.05); the statistical significance of the layers and ends was independent. ML = mammillary layer; PL = palisade layer; VCL = vertical crystal layer; QBPs = quantity of bubble pores; AABPs = average areas of bubble pores; ASBPs = sum of the area of bubble pores; ARBE = ASBPs/the area of the eggshell in each image; and APBPs = average perimeter of bubble pores.

**Table 4 animals-14-01496-t004:** Differences in eggshell chemical components between Dwarf Layer-White and Guinea fowl.

Traits	Dwarf Layer-White	Guinea Fowl
Organic matter	3.15 ± 0.20 ^B^	4.10 ± 0.51 ^A^
Inorganic matter	96.85 ± 0.20 ^A^	95.90 ± 0.51 ^B^
Ca	32.45 ± 1.83 ^b^	34.52 ± 0.58 ^a^
P	0.12 ± 0.02 ^A^	0.07 ± 0.02 ^B^
Mg	0.75 ± 0.21	0.63 ± 0.12

In the same row, values with no letter mean no significant difference (*p* > 0.05), while values with different lowercase letter superscripts indicate significant differences (*p* < 0.05), while values with different capital letter superscripts indicate significant differences (*p* < 0.01).

**Table 5 animals-14-01496-t005:** Correlations between eggshell quality and ultrastructure traits of Dwarf Layer-White and Guinea fowl ^1^.

Traits ^2^	ESS	EST	ESMT	MT	ET	MAN	QBPs	AABPs	ASBPs	ARBE
ESS	1	0.783 **	0.124	0.173	0.548 **	0.111	0.131	0.258	0.188	0.246
EST	0.652 **	1	0.140	0.769 **	0.911 **	−0.053	0.216	0.237	0.206	0.199
ESMT	0.161	0.090	1	−0.135	−0.104	0.063	0.240	−0.299	−0.131	0.022
MT	0.286	0.611 **	−0.008	1	0.257	−0.423 **	0.041	0.325	0.313	0.345
ET	0.534 *	0.829 **	−0.015	0.347	1	−0.030	0.042	0.113	0.040	0.024
MAN	0.175	−0.171	0.094	−0.551 **	0.019	1	0.356 *	−0.463 *	0.592 *	0.465 *
QBPs	−0.455 *	−0.210	−0.111	−0.074	−0.204	−0.300	1	−0.348 *	0.655 **	0.529 **
AABPs	0.241	0.225	−0.195	0.049	0.260	0.363	−0.629 **	1	0.478 *	0.284
ASBPs	−0.084	0.146	−0.078	0.103	0.329	−0.357	0.501 *	0.439 *	1	0.712 **
ARBE	−0.256	−0.359	−0.269	−0.203	−0.143	0.196	0.417 *	0.308	0.398 *	1

^1^ Correlations of Dwarf Layer-White above diagonal and correlations of Guinea fowl below diagonal. ^2^ ESS = eggshell strength; EST = eggshell thickness; ESMT = eggshell membrane thickness; MT = mammillary layer thickness; ET = effective layer thickness; MAN = mammillary knob number in image; QBPs = quantity of bubble pores; AABPs = average areas of bubble pores; ASBPs = sum of the area of bubble pores; and ARBE = ASBPs/the area of the eggshell in each image. * *p* ≤ 0.05; ** *p* ≤ 0.01.

## Data Availability

The data supporting the results of this article will be made available by the authors on request.

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
