# Peer review of "Comparative Analysis of the Ultrastructure, Bubble Pores, and Composition of Eggshells of Dwarf Layer-White and Guinea Fowl"

_animals, 2024, doi:10.3390/ani14101496_

Round 1

Reviewer 1 Report

Comments and Suggestions for Authors

Please see the attached pdf with comments

Reviewer 2 Report

Comments and Suggestions for Authors

The manuscript submitted for review addresses the ultrastructure of eggshells of two species and is in line with current research trends in the field of animal sciences. Before proceeding further, however, it requires revisions, a list of which is provided below:

 1.   Simple summary: needs rewording. The word "quality" appears 3 times in the first sentence.

2.    The abstract is far too long, according to the requirements of the journal, it can be a maximum of 200 words. Please indicate in it the purpose of the work, the number of eggs analyzed, the parameters of the analysis, the main results and conclusions.

Introduction:

This chapter is written correctly, but I suggest supplementing it with eggshell pigments. Their role in shaping the physical and chemical characteristics of shells has already been described for both wild and domesticated birds (10.3390/ani11051204 ; 10.3390/ani10020264; 10.1007/s10336-010-0551-7).

Material and methods

Line 124-incorrect way of citation

Please indicate the general composition of the feed mixture- was it uniform? What was the calcium content of the feed? What % laying rate was observed in the birds at the time the study was conducted? These are very important elements for the results obtained.

Line 148- please replace acute with sharp end

In the description of the ultrastructure analysis, please indicate the type of SE/BSE detector?

Samples were glued to carbon tape? Were other stablization methods used especially for cross sections? Please indicate the conditions in the microscope chamber (vacuum)

Figure 1 should be in the results section rather than M&M

Line 211- incorrect citation

If AAS was used please indicate what equipment was used for mineralization, what dilutions were used, were reference materials used? If so, which ones?

Also, after boiling in NaOH, determination of Na content seems to be unwarranted

RESULTS and discussion

Line 321- please verify ion recording

The discussion needs improvement and the use of more literature data, especially since there have been quite a few recent reports on shell ultrastructure. The problem is the limited amount of data on guinea fowl, but the authors indicate that the relationships are similar to those in hens, so this section can be improved.

The references need significant improvement-check them carefully and be sure to add DOI numbers.

Round 2

Reviewer 2 Report

Comments and Suggestions for Authors

The authors addressed all my corrections. I believe that in its present form the article is suitable for publication.